# Impact of Hyponatremia on COVID-19-Related Outcomes: A Retrospective Analysis

**DOI:** 10.3390/biomedicines12091997

**Published:** 2024-09-02

**Authors:** Pedro Maciel de Toledo Piza, Victor Muniz de Freitas, Isabella Aguiar-Brito, Barbara Monique Calsolari-Oliveira, Érika Bevilaqua Rangel

**Affiliations:** 1Paulista School of Medicine, Federal University of São Paulo, São Paulo 04023-062, SP, Brazil; pedro.piza@unifesp.br (P.M.d.T.P.); muniz.victor@unifesp.br (V.M.d.F.); isabella.brito@unifesp.br (I.A.-B.); barbara.calsolari@unifesp.br (B.M.C.-O.); 2Department of Medicine, Nephrology Division, Federal University of São Paulo, São Paulo 04038-031, SP, Brazil; 3Instituto Israelita de Ensino e Pesquisa Albert Einstein, Hospital Israelita Albert Einstein, São Paulo 05652-900, SP, Brazil

**Keywords:** sodium, hyponatremia, COVID-19, outcomes

## Abstract

Background: Sodium disturbances are observed in one-third of patients with COVID-19 and result from multifaceted mechanisms. Notably, hyponatremia is associated with disease progression and mortality. Aim: We aimed to analyze the impact of hyponatremia on COVID-19 outcomes and its correlation with clinical and laboratory parameters during the first wave. Methods: We evaluated the sodium levels of 558 patients with COVID-19 between 21 March 2020, and 31 July 2020, at a single center. We performed linear regression analyses to explore the correlation of sodium levels with COVID-19-related outcomes, demographic data, signs and symptoms, and laboratory parameters. Next, we conducted Pearson correlation analyses. A *p*-value < 0.05 was considered significant. Results: Hyponatremia was found in 35.3% of hospitalized patients with COVID-19. This was associated with the need for intensive care transfer (B = −1.210, *p* = 0.009) and invasive mechanical ventilation (B = −1.063, *p* = 0.032). Hyponatremia was frequently found in oncologic patients (*p* = 0.002) and solid organ transplant recipients (*p* < 0.001). Sodium was positively associated with diastolic blood pressure (*p* = 0.041) and productive cough (*p* = 0.022) and negatively associated with dry cough (*p* = 0.032), anorexia (*p* = 0.004), and nausea/vomiting (*p* = 0.007). Regarding the correlation of sodium levels with other laboratory parameters, we observed a positive correlation with hematocrit (*p* = 0.011), lymphocytes (*p* = 0.010), pCO_2_ (*p* < 0.0001), bicarbonate (*p* = 0.0001), and base excess (*p* = 0.008) and a negative correlation with the neutrophil-to-lymphocyte ratio (*p* = 0.009), the platelet-to-lymphocyte ratio (*p* = 0.033), and arterial blood glucose (*p* = 0.016). Conclusions: Hyponatremia is a risk factor for adverse outcomes in COVID-19 patients. It is associated with demographic data and clinical and laboratory parameters. Therefore, hyponatremia is an important tool for risk stratification in COVID-19 patients.

## 1. Introduction

At the beginning of the COVID-19 (coronavirus disease 2019) pandemic, caused by the Severe Acute Respiratory Syndrome Coronavirus-2 (SARS-CoV-2), the most prevalent symptoms were mild and included fever and cough. However, approximately 20% of patients required hospitalization in the pre-vaccination era [1].

The SARS-CoV-2 virus exerts its effects primarily through the angiotensin-converting enzyme 2 (ACE2) receptor, which is highly expressed on the surface of various cells, including those in the heart, kidneys, lungs, and endothelial cells [2]. The pathogenicity of SARS-CoV-2 involves a combination of direct viral cytotoxicity and a dysregulated immune response that includes severe inflammation and thrombosis [3]. The interplay of these mechanisms results in significant morbidity and mortality, especially in patients with pre-existing comorbidities and in severe cases of COVID-19.

In low-income and middle-income countries, the impact of COVID-19 has been profound and multifaceted, affecting health systems, economies, and social structures. In Brazil, for example, the in-hospital mortality rate was 39% overall, 59% among those admitted to the intensive care unit (ICU), 80% among those who required invasive mechanical ventilation (IMV) [4], and 72.5% among those who developed acute kidney injury (AKI) and required renal replacement therapy (RRT) [5]. 

The utilization of machine learning approaches, such as the RAPID RISK COVID model developed at our institution, has allowed the identification of patients at higher risk of hospitalization [6]. This model identified key predictors, including lower oxyhemoglobin saturation by pulse oximetry (SpO_2_), higher respiratory rate, fever, higher heart rate, and lower blood pressure levels, associated with age, male sex, and underlying conditions such as diabetes mellitus (DM) and hypertension.

Not only are signs and symptoms important, but specific tomography findings indicative of lung involvement, underlying comorbidities, and diverse laboratory test abnormalities have also been demonstrated as predictors of COVID-19 progression and mortality [7,8]. These laboratory parameters include inflammatory markers (C-reactive protein), cellular injury markers (lactate dehydrogenase), coagulation abnormalities (high levels of D-dimer, thrombocytopenia, excessive fibrin polymerization, and thrombosis), cardiac dysfunction markers (high levels of troponin and brain natriuretic peptides), alterations in white blood cell counts (mainly lymphopenia and neutrophilia), liver injury markers (elevations in aspartate aminotransferase and alanine aminotransferase), and renal dysfunction.

In line with these findings, the association between serum sodium levels at hospital admission and mortality demonstrated a U-shaped pattern, with hyponatremia being more frequently observed (35.5%) compared to hypernatremia (9.1%) [9]. Hyponatremia has been identified as a marker of COVID-19-related adverse outcomes such as mortality, need for ICU transfer, and IMV [10]. In a meta-analysis of 81 studies (*n* = 850,222 patients) from the pre-COVID-19 era, hyponatremia was found in 17.4% of inpatient settings and was independently associated with overall mortality (relative risk [RR] = 2.60) [11]. Hyponatremia was also associated with an increased risk of mortality in patients with myocardial infarction (RR  =  2.83), heart failure (RR  =  2.47), cirrhosis (RR  =  3.34), pulmonary infections (RR  =  2.49), mixed diseases (RR  =  2.59), and in hospitalized patients (RR  =  2.48).

Therefore, we aimed to investigate the impact of sodium levels, particularly hyponatremia at admission, in a cohort of hospitalized patients following a COVID-19 diagnosis during the first wave of the pandemic. Specifically, we examined the association between sodium levels and various demographic, clinical, and laboratory parameters, as well as outcomes such as mortality, transfer to the ICU, need for oxygen (O_2_), IMV, and RRT.

## 2. Materials and Methods

### 2.1. Study Population

We reviewed data from all patients referred to the Paulista School of Medicine at Hospital São Paulo, a tertiary university hospital at the Federal University of São Paulo (EPM-UNIFESP), SP, Brazil, between 21 March 2020 and 31 July 2020. These patients were admitted to the hospital with respiratory syndrome during the first wave of the COVID-19 pandemic. A total of 578 patients were hospitalized. Exclusion criteria were patients under 18 years old and the absence of sodium measurement within the first 24 h of hospital admission and/or ICU admission. Due to the high burden of COVID-19 during the first wave of the pandemic, the majority of patients were promptly transferred to the ICU. However, some patients were intubated and ventilated or maintained on non-invasive ventilation in the emergency room and even within the wards before being transferred to the ICU. As a result, sodium measurements were not available for 20 patients at the time of hospital/ICU admission, so they were excluded from the analysis. Consequently, 558 patients were included in the study.

### 2.2. Assessment

We evaluated medical history; clinical parameters including vital signs such as blood pressure, respiratory rate, heart rate, temperature, and oxyhemoglobin saturation by pulse oximetry (SpO_2_); demographic data; pre-existing comorbidities; and laboratory parameters on hospital admission. When data were missing in more than 45% of the samples, those samples were excluded from the analyses, as shown in Appendix A. Hospitalization was based on medical decisions, in particular, when SpO_2_ was lower than 94% and respiratory rate was greater than 24 bpm. After hospitalization, more than 90% of the individuals tested positive for SARS-CoV-2 using RT-PCR from nasopharyngeal samples.

All data were registered in the electronic health record (EHR) and manually imputed by the researchers. Discrepancies were solved by two of the investigators (P.M.T.P. and V.M.F.) after reviewing the entirety of the patient’s chart. When discrepancies could not be solved or data were not registered, it was marked as “not available”.

The Ethics Committee from the Federal University of São Paulo approved the study (CAAE: 41400720.7.0000.5505). All the methods were performed following guidelines and regulations. In addition, this study was carried out under the Declaration of Helsinki. The requirement for informed consent was waived by the Ethics Committee because our study used anonymized data for analysis.

### 2.3. Statistical Analyses

The primary objective was to evaluate the association between serum sodium levels and adverse outcomes (death, ICU admission, need for O_2_ therapy, need for IMV, and need for dialysis) in patients admitted to Hospital São Paulo, SP, Brazil. Hyponatremia was defined by serum values less than 135 mEq/L, and hypernatremia by values greater than 145 mEq/L.

The secondary objectives were to relate the demographic and laboratory parameters, as well as the symptoms of these patients, to sodium levels upon admission. To investigate how sodium levels relate to outcomes and other variables, we performed a univariate analysis using linear regression to examine the relationship between sodium levels and outcomes, demographic data, symptoms, and laboratory data. When the *p*-value was ≤0.1, the variables were simultaneously introduced into a multivariate linear regression model. The results were expressed using the B value, which determines the relationship between serum sodium levels and the variable in question. Positive B values indicate that sodium levels are positively related to the variable, while negative B values indicate a negative relationship.

If laboratory data showed a statistically significant relationship with serum sodium levels, this relationship was represented in a scatter plot and analyzed using Pearson correlation.

Numerical variables were described using either their mean ± standard deviation or the median and interquartile range, as appropriate. For categorical variables, frequencies and percentages were reported. Sodium distribution was also evaluated using a histogram graphic.

The confidence interval (CI) was 95%, and a *p*-value < 0.05 was considered statistically significant in all analyses. We analyzed the data using IBM^®^ SPSS (Statistical Product and Services Solutions, version 18.0, SPSS Inc, Chicago, IL, USA).

## 3. Results

We evaluated 558 hospitalized patients with COVID-19. The average sodium level was 135.95 ± 5.48 mEq/L, with a median level of 136 [133; 139] mEq/L. Figure 1 depicts the sodium distribution at admission. In our cohort, 197 patients (35.3%) had sodium levels below 135 mEq/L while 15 patients (2.7%) had sodium values above 145 mEq/L.

Sodium levels were independently and negatively associated with adverse COVID-19-related outcomes, particularly the need for ICU transfer and IMV (Table 1).

To further investigate the demographic data associated with sodium levels, we performed a linear regression analysis (Table 2). We observed that the presence of neoplasia and the use of immunosuppressive therapy were negatively associated with sodium levels by both univariate and multivariate analyses.

When we evaluated the symptoms and signs at admission, we found that dry cough, anorexia, nausea/vomiting, and diastolic blood pressure values were negatively associated with sodium levels in univariate analyses (Table 3). In contrast, productive cough was positively associated with sodium levels. In multivariate analysis, only nausea/vomiting and anorexia remained significantly associated with lower sodium levels.

In the univariate analysis, several laboratory parameters positively predicted sodium levels, including pCO_2_, bicarbonate, base excess, hematocrit, and lymphocytes (Table 4). Conversely, arterial blood glucose levels, the neutrophil-to-lymphocyte ratio, and the platelet-to-lymphocyte ratio negatively predicted sodium levels. In the multivariate analysis, hematocrit and bicarbonate remained positive predictors, while arterial blood glucose continued to be a negative predictor, and the neutrophil-to-lymphocyte ratio also emerged as a negative predictor of sodium levels.

In Figure 2A–I, we illustrate the correlation between sodium values and various laboratory and clinical parameters. We found a positive correlation between sodium levels and pCO_2_, bicarbonate, base excess, hematocrit, lymphocytes, and diastolic blood pressure, as shown in blue. Conversely, we observed a negative correlation between sodium levels and the neutrophil-to-lymphocyte ratio, the platelet-to-lymphocyte ratio, and arterial blood glucose, as shown in red.

## 4. Discussion

In our study, we found that sodium abnormalities were present in almost 40% of the patients admitted during the first wave of the COVID-19 pandemic. Hyponatremia was the most prevalent sodium disturbance at admission. Various demographic, clinical, and laboratory parameters were associated with hyponatremia, particularly in patients with immunosuppressive conditions, symptoms, and signs such as dry cough, anorexia, nausea/vomiting, and low diastolic blood pressure. Additionally, several laboratory parameters predicted lower sodium levels, including blood gas parameters, glucose, lymphocyte count, and hematocrit. We also observed that hyponatremia was associated with COVID-19 severity, including the need for ICU transfer and IMV.

The pathophysiologic mechanisms associated with hyponatremia are complex and largely dependent on the volume status [12]. Hyponatremia can occur in hypovolemic, euvolemic, and hypervolemic conditions. Hypovolemic hyponatremia occurs when both sodium and water are lost via renal and extra-renal routes, particularly through the gastrointestinal, respiratory, and skin routes. Euvolemic hyponatremia is typically observed in the syndrome of inappropriate antidiuretic hormone secretion (SIHAD). Hypervolemic hyponatremia is associated with kidney, heart, or liver dysfunction. Symptoms and signs of hyponatremia vary from mild and nonspecific, such as weakness or nausea, to severe and life-threatening, such as seizures and coma [12].

The incidence of hyponatremia in the COVID-19 setting has been reported to range from 9.9% to 60% [13], consistent with our findings (35.3%). In a systematic review and meta-analysis of 23 studies, hyponatremia was associated with increased mortality (odds ratio [OR] = 1.97), ICU admission (OR = 1.91), and need for IMV (OR = 2.04) in COVID-19 patients [14]. Except for mortality, our findings were consistent with those reported in the literature. The adverse outcomes in COVID-19 patients admitted with hyponatremia may be attributed to more severe pulmonary lesions identified on thoracic computed tomography scans upon admission in this population [10].

Regarding the mechanisms of hyponatremia in COVID-19 patients, SIADH was the most frequently reported, followed by adrenal insufficiency and finally hypovolemic hyponatremia due to gastrointestinal losses [14]. Other causes include volume and sodium depletion due to a combination of diuretics and poor oral intake [15]. Although we did not evaluate the volume status of the COVID-19 patients, we speculate that hyponatremia in our study may be explained by poor water and salt intake, as anorexia and low diastolic blood pressure were independently associated with hyponatremia. Additionally, our patients frequently experienced nausea and vomiting, which could have led to hypovolemic hyponatremia. The presence of dry cough could also indicate more severe damage to the lungs and the need for IMV, associated with euvolemic hyponatremia due to SIADH.

In line with these findings, the arterial blood gas analyses in our COVID-19 patients demonstrated a positive correlation between sodium and pCO_2_, suggesting hyperventilating as a compensatory mechanism for metabolic acidosis [16] due to tissue hypoxia or lactic acidosis in those patients with lower sodium levels. Importantly, the positive correlation of sodium and base excess indicates that a negative base excess in those patients with lower sodium levels exhibits a deficit of base (bicarbonate) in the blood. This is typically seen in conditions such as metabolic acidosis, where there is an accumulation of acids (e.g., lactic acid, ketoacids) or a loss of bicarbonate. In COVID-19 patients, a negative base excess might indicate metabolic acidosis, which can occur due to tissue hypoxia, sepsis, or renal dysfunction [16,17].

In the COVID-19 context, sodium levels were found to be inversely correlated with the pro-inflammatory cytokine IL-6 and positively correlated with the PaO_2_/FiO_2_ ratio [18], suggesting that hyponatremic patients with COVID-19 exhibited increased inflammation and reduced ventilator capacity. Whether IL-6 released by macrophages and monocytes in the inflammatory response can directly stimulate thirst and water intake—as well as increase vasopressin secretion by osmoreceptors in neurons from the supraoptic and paraventricular nuclei in the hypothalamus, consequently leading to SIADH [19]—remains to be elucidated in the COVID-19 setting.

We also observed that inflammatory markers, such as lymphopenia and higher neutrophil-to-lymphocyte ratios and platelet-to-lymphocyte ratios, were predictors of hyponatremia. The mechanisms leading to lymphopenia in COVID-19 involve several factors [2,20,21,22,23,24]. These factors include direct viral infection, as the SARS-CoV-2 can directly infect lymphocytes—particularly T cells—by binding to the ACE2 receptor, leading to their destruction; cytokine storm, which is characterized by high levels of inflammatory cytokines like IL-6 and TNF-α and can lead to lymphocyte apoptosis; bone marrow suppression, as the inflammatory cytokines and the viral infection itself can suppress bone marrow function, reducing lymphocyte production; the redistribution of lymphocytes, as during severe infection, lymphocytes can be redistributed from the peripheral blood to tissues, including the lungs and lymphoid organs, in response to the infection; the exhaustion and senescence of lymphocytes, as the prolonged activation of the immune system can lead to the exhaustion and functional impairment of lymphocytes, resulting in decreased proliferation and increased cell death; and an immunosuppressive environment, as COVID-19 can create an immunosuppressive environment, with decreased regulatory T cells and increased myeloid-derived suppressor cells that suppress lymphocyte function and proliferation. These mechanisms collectively contribute to the lymphopenia observed in many COVID-19 patients and are often associated with worse clinical outcomes, including increased severity of disease and higher mortality rates [25].

Neutrophils, in turn, play a significant role in the innate immune response to COVID-19 and can influence disease outcomes in several ways [26]. They are recruited to the site of infection where they engulf and destroy pathogens through phagocytosis and the release of antimicrobial substances. Similar neutrophilia and lymphopenia and an elevated neutrophil-to-lymphocyte ratio have been observed in severe cases of COVID-19, indicating a more pronounced inflammatory response that can contribute to tissue damage and disease severity [27,28]. Additionally, neutrophils can release extracellular traps (NETs) to trap and kill pathogens [26]. However, excessive NET formation can also lead to tissue damage and exacerbate inflammation, potentially worsening the severity of COVID-19. Neutrophils can contribute to the cytokine storm through cytokine release, further exacerbating the inflammatory cascade and tissue damage. In severe COVID-19, there may also be the dysfunction or exhaustion of neutrophils, impairing their ability to effectively combat the virus and contributing to prolonged inflammation and tissue injury.

Platelets, on the other hand, play roles not only in hemostasis but also in inflammation and immune response modulation. They can contribute to the inflammatory response seen in COVID-19 through interactions with immune cells and endothelial cells [29]. The platelet-to-lymphocyte ratio reflects the balance between the inflammatory response (represented by platelets) and the immune response (represented by lymphocytes) in COVID-19 patients. A higher ratio has been associated with more severe disease and poorer outcomes in COVID-19 patients, as it correlates with the degree of inflammation and immune dysregulation observed in severe cases [30].

Inflammation and immune dysregulation can also be associated with lower hematocrit levels in the COVID-19 setting [31], as we also observed. SARS-CoV-2 can directly damage the red blood cells, altering their size, rigidity, and distribution width and impairing their functionality [32].

Glucose levels can significantly impact COVID-19 outcomes due to several inter-related factors. Hyperglycemia can impair both immune and adaptative immune responses, contributing to chronic inflammation [33], which can exacerbate the inflammatory response observed in severe cases of COVID-19 [34]. Additionally, hyperglycemia can increase ACE2 glycation and TMRSS-2 expression, resulting in increased viral binding and S-protein priming, ultimately facilitating the spread of SARS-CoV-2 [35]. This increased inflammation can result in more severe symptoms and complications in COVID-19 patients.

Furthermore, a high prevalence of aging, cardiovascular disease, obesity, and hypertension among individuals with diabetes mellitus poses additional risk factors for severe COVID-19 outcomes [34]. Moreover, COVID-19 itself can induce insulin resistance and adipose tissue infectivity [36], as well as cause direct damage to β-cells [37].

Importantly, glucose levels at admission are associated with COVID-19-related adverse outcomes in patients both with [38] and without [39] pre-existing diabetes mellitus, leading to unstable glucose levels throughout the illness [40,41]. These conditions are linked to poorer prognosis and higher mortality rates among COVID-19 patients.

Hyperglycemia can be associated with hyponatremia due to several mechanisms [42,43]. It promotes osmotic diuresis, resulting in the loss of water and sodium. Hyperglycemia also has a dilutional effect, as glucose is an osmotically active substance, causing an increase in blood osmolality. To maintain osmotic balance, water shifts from cells into the extracellular space. This dilutional effect can lead to a relative decrease in sodium concentration in the blood, even if the total amount of sodium is normal or slightly decreased. Both osmotic diuresis and the dilutional effect can lead to dehydration and hypovolemia. In response, the body may retain sodium and water in an attempt to restore fluid balance, but the relative excess of water compared to sodium can result in hyponatremia.

Finally, solid organ recipients [44] and oncology patients [45] have impaired immune system function due to their treatments, placing them at a heightened risk of infection and resulting in worse outcomes in the context of COVID-19, as we observed in our study. Immunosuppressive drugs [46] and chemotherapy [47] also contribute to gastrointestinal effects, including nausea, vomiting, anorexia, and diarrhea, which may lead to electrolyte imbalances such as lower sodium levels. These patients also exhibit a higher prevalence of comorbidities and tend to be older than the general population, further increasing their susceptibility to COVID-19 progression and mortality [48,49].

Our study has several limitations. Firstly, we were unable to ascertain the volumetric status, which prevented us from determining the precise cause of sodium disturbances. The use of other medications that could potentially interfere with electrolyte balance, such as diuretics and corticosteroids, as well as the assessment of adrenal and thyroid function, were not available. Secondly, longitudinal laboratory parameters were not evaluated, which could have provided further insights into the correlation between sodium levels and these parameters. Thirdly, we found a weak correlation between sodium levels and laboratory parameters. This can be explained by non-linear relationships, the presence of outliers, the range of data, and sample size. Furthermore, our study was conducted at a single center and was retrospective in nature, which may impact the thoroughness of our analyses due to incomplete data. Finally, our findings should be validated in the context of the use of drugs with demonstrated efficacy against COVID-19 and vaccination.

## 5. Conclusions

In conclusion, clinical and demographic parameters can predict sodium values upon admission. The early identification of factors that predict COVID-19 progression aids in therapeutic decision-making and patient flow management. Identifying predictors of poor outcomes in COVID-19 is crucial for effective risk stratification. Additionally, our study provides further biological insights into disease progression. Thus, our findings contribute to resource planning and allocation and offer valuable data for developing machine learning models in future pandemic waves of COVID-19 and other respiratory viruses in general.

## Figures and Tables

**Figure 1 biomedicines-12-01997-f001:**
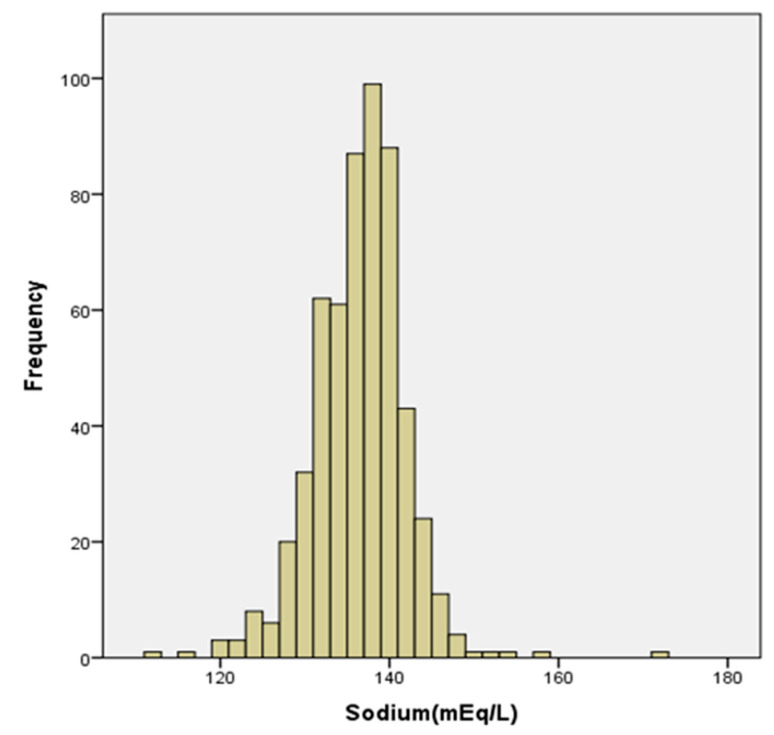
Sodium value distribution at admission (*n* = 558).

**Figure 2 biomedicines-12-01997-f002:**
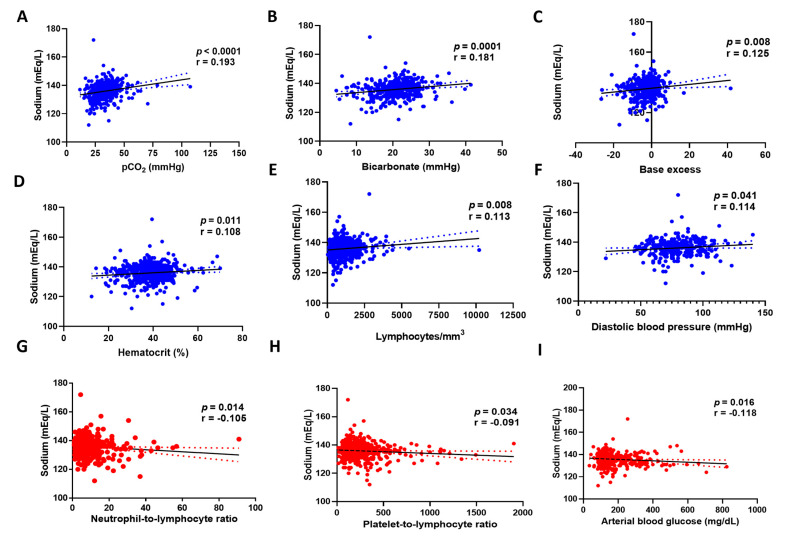
Pearson correlation between sodium levels and (**A**) pCO_2_, (**B**) bicarbonate, (**C**) base excess, (**D**) hematocrit, (**E**) lymphocytes, (**F**) diastolic blood pressure, (**G**) neutrophil-to-lymphocyte ratio, (**H**) platelet-to-lymphocyte ratio, and (**I**) arterial blood glucose. *N* = 558 patients.

**Table 1 biomedicines-12-01997-t001:** Linear regression analyses of outcomes and sodium levels.

Variables	*n* = 558	Univariate Analysis	Multivariate Analysis
Mortality (*n*, %)	172 (30.8%)	−0.913 (−1.898–0.071, *p* = 0.069)	−0.046 (−1.805–0.716, *p* = 0.397)
ICU (*n*, %)	263 (47.1%)	**−1.210 (−2.118–−0.303, *p* = 0.009)**	−0.112 (−2.454–0.008, *p* = 0.051)
O_2_ (*n*, %)	482 (86.4%)	0.132 (−1.211–1.475, *p* = 0.847)	
IMV (*n*, %)	179 (32.1%)	**−1.063 (−1.036–−0.90, *p* = 0.032)**	0.012 (−1.380–1.651, *p* = 0.861)
RRT (*n*, %)	123 (22.0%)	−1.037 (−2.133–0.58, *p* = 0.063)	−0.016 (−1.541–1.118, *p* = 0.755)

ICU = Intensive Care Unit; O_2_ = Oxygen; IMV = Invasive Mechanical Ventilation; RRT = Renal Replacement Therapy. The **bold** values indicate statistical significance.

**Table 2 biomedicines-12-01997-t002:** Linear regression analyses of demographic data and sodium levels.

Variables	*n* = 558	Univariate Analysis	Multivariate Analysis
Age (years)	59.1 ± 16.1	0.008 (−0.020–0.037, *p* = 0.567)	
Male (*n*, %)	317 (56.8%)	0.465 (−0.455–1.385, *p* = 0.321)	
Obesity (*n*, %)	42 (11.2%)	−0.237 (−2.068–1.593, *p* = 0.799)	
Hypertension (*n*, %)	177 (47.5%)	0.124 (−1.036–1.285, *p* = 0.833)	
COPD (*n*, %)	9 (2.4%)	−0.696 (−4.467–3.076, *p* = 0.717)	
Heart disease (*n*, %)	52 (13.9%)	0.830 (−0.841–2.502, *p* = 0.329)	
Neoplasia (*n*, %)	24 (6.4%)	**−3.665 (−5.995–−1.335, *p* = 0.002)**	**−0.164 (−6.081–−1.514, *p* = 0.001)**
Respiratory disease (*n*, %)	9 (2.4%)	2.492 (−1.271–6.256, *p* = 0.194)	
Kidney disease (*n*, %)	50 (13.4%)	−0.717 (−2.414–0.980, *p* = 0.407)	
Transplant (*n*, %)	54 (14.4%)	−1.313 (−2.952–0.326, *p* = 0.116)	
Asthma (*n*, %)	12 (3.2%)	−0.328 (−3.608–2.952, *p* = 0.844)	
Diabetes mellitus (*n*, %)	105 (28.1%)	−0.829 (−2.113–0.455, *p* = 0.205)	
Cerebrovascular disease (*n*, %)	15 (4%)	0.728 (−2.217–3.673, *p* = 0.627)	
Immunosuppressive therapy (*n*, %)	44 (11.8%)	**−3.520 (−5.278–−1.762, *p* < 0.001)**	**−0.204 (−5.324–−1.850, *p* < 0.001)**
Tuberculosis (*n*, %)	2 (0.4%)	2.110 (−5.813–10.033, *p* = 0.601)	
Pregnancy (*n*, %)	2 (0.4%)	−2.414 (−10.336–5.508, *p* = 0.549)	
Smoking (*n*, %)	32 (8.6%)	−0.131 (−2.198–1.936, *p* = 0.901)	

Age represents mean ± SD. COPD = chronic obstructive pulmonary disease. The **bold** values indicate statistical significance.

**Table 3 biomedicines-12-01997-t003:** Linear regression analyses of signs and symptoms and sodium levels.

Variables	*n* = 558	Univariate Analysis	Multivariate Analysis
Fever (*n*, %)	213 (57%)	−0.381 (−1.548–0.786, *p* = 0.521)	
Fatigue (*n*, %)	139 (37.2%)	−1.010 (−2.202–0.181, *p* = 0.096)	−0.565 (−1.775–0.646, *p* = 0.360)
Sneezing (*n*, %)	12 (3.2%)	−0.156 (−3.436–3.124, *p* = 0.926)	
Dry cough (*n*, %)	200 (53.5%)	**−1.260 (−2.441–−0.108, *p* = 0.032)**	−0.494 (−1.748–0.760, *p* = 0.439)
Productive cough (*n*, %)	56 (15%)	**1.881 (0.272–3.489, *p* = 0.022)**	1.444 (−0.284–3.173, *p* = 0.101)
Rhinorrhea (*n*, %)	33 (8.8%)	−0.939 (−1.096–2.975, *p* = 0.365)	
Sore throat (*n*, %)	23 (6.1%)	−0.404 (−2.810–2.002, *p* = 0.741)	
Diarrhea (*n*, %)	72 (19.3%)	−1.081 (−2.543–0.380, *p* = 0.147)	
Dyspnea (*n*, %)	228 (61%)	−0.027 (−1.212–1.157, *p* = 0.964)	
Anorexia (*n*, %)	67 (17.9%)	**−2.225 (−3.715–−0.715, *p* = 0.004)**	**−1.703 (−3.255–−0.151, *p* = 0.032)**
Headache (*n*, %)	63 (16.8%)	−0.721 (−2.264–0.822, *p* = 0.359)	
Myalgia (*n*, %)	99 (26.5%)	−0.947 (−2.254–0.361, *p* = 0.361)	
Nausea/vomiting (*n*, %)	71 (19%)	**−2.028 (−3.490–−0.566, *p* = 0.007)**	**−1.627 (−3.089–−0.165, *p* = 0.029)**
Wheezing (*n*, %)	6 (1.6%)	−0.408 (−5.008–4.193, *p* = 0.862)	
Chest pain (*n*, %)	44 (11.8%)	−0.529 (−2.325–1.267, *p* = 0.563)	
Abdominal pain (*n*, %)	23 (6.1%)	−1.006 (−3.410–1.398, *p* = 0.411)	
Anosmia (*n*, %)	57 (15.2%)	0.324 (−1.284–1.932, *p* = 0.692)	
Dysgeusia (*n*, %)	54 (14.5%)	−0.323 (−1.968–1.322, *p* = 0.700)	
Chills (*n*, %)	24 (6.4%)	0.551 (−1.807–2.909, *p* = 0.646)	
Temperature (°C)	36.6 ± 0.7	0.491 (−0.437–1.418, *p* = 0.299)	
MBP (mmHg)	95 [84; 106]	0.034 (−0.003–0.070, *p* = 0.070)	−0.028 (−0.149–0.092, *p* = 0.646)
SBP (mmHg)	128 [113.5; 143.5]	0.017 (−0.010–0.045, *p* = 0.218)	
DBP (mmHg)	78 [69; 90]	**0.040 (0.002–0.079, *p* = 0.041)**	0.069 (−0.059–0.197, *p* = 0.290)
HR (bpm)	95 [82; 110]	−0.016 (−0.049–0.017, *p* = 0.345)	
SpO_2_ (%)	91.1 ± 6.71	−0.017 (−0.107–0.074, *p* = 0.717)	
Shock Index	0.75 ± 0.3	−1.568 (−4.001–0.865, *p* = 0.206)	
RR (bpm)	24 [20.5; 29]	0.040 (−0.062–0.143, *p* = 0.779)	

MBP = mean blood pressure; SBP = systolic blood pressure; DBP = diastolic blood pressure; HR = heart rate (in bpm, beats per minute); SpO_2_ = peripheral capillary oxygen saturation; RR = respiratory rate (in bpm, breaths per minute). Shock index is calculated as heart rate/systolic blood pressure. Temperature, SpO_2_, and shock index are presented as means ± SD. MBP, SBP, DBP, HR, and RR are presented as medians and IQR. The **bold** values indicate statistical significance.

**Table 4 biomedicines-12-01997-t004:** Linear regression analyses of laboratory parameters and sodium levels.

Variables	*n* = 558	Univariate Analysis	Multivariate Analysis
Hemoglobin (g/dL)	13 ± 2.6	0.115 (−0.058–0.288, *p* = 0.193)	
Hematocrit (%)	38.8 ± 7.4	**0.080 (0.018–0.141, *p* = 0.011)**	**0.110 (0.04-0–0.180, *p* = 0.002)**
Leucocytes (mm^3^)	7880 [5412.5; 10,775]	−1.792 × 10^−5^ (0.000–0.000, *p* = 0.703)	
Neutrophils (mm^3^)	6061 [4019.5; 8658.5]	−4.083 × 10^−5^ (0.000–0.000, *p* = 0.481)	
Eosinophils (mm^3^)	0 [0; 32]	0.001 (−0.003–0.005, *p* = 0.657)	
Basophils (mm^3^)	8 [0; 22]	0.012 (−0.010–0.034, *p* = 0.281)	
Lymphocytes (mm^3^)	1131.47 ± 821.2	**0.001 (0.000–0.001, *p* = 0.010)**	0.000 (0.000–0.001, *p* = 0.259)
Atypical lymphocytes (mm^3^)	0 [0; 0]	0.000 (−0.002–0.002, *p* = 0.873)	
Monocytes (mm^3^)	432.5 [276; 649]	2.625 × 10^−5^ (−0.001–0.001, *p* = 0.964)	
Platelets (×10^3^, mm^3^)	190 [144; 251]	3.540 × 10^−6^ (0.000–0.000, *p* = 0.183)	
Neutrophil-to-lymphocyte ratio	5.5 [3.8; 10.2]	**−0.074 (−0.129–−0.019, *p* = 0.009)**	**−0.104 (−0.189–−0.019, *p* = 0.016)**
Platelet-to-lymphocyte ratio	199.7 [127.7; 310]	**−0.002 (−0.005–0.000, *p* = 0.033)**	0.002 (−0.001–0.006, *p* = 0.196)
D-dimer (µg/L)	1.4 [0.8; 2.5]	0.048 (−0.115–0.211, *p* = 0.564)	
Lactate (mmol/L)	13.2 [10; 19]	0.004 (−0.045–0.052, *p* = 0.876)	
CRP (mg/dL)	100.6 [51.8; 186.7]	−0.004 (−0.009–0.001, *p* = 0.106)	
LDH (U/L)	415.65 ± 210.9	−0.002 (−0.005–0.001, *p* = 0.126)	
Creatinine (mg/dL)	1.1 [0.8; 1.9]	−0.044 (−0.243–0.155, *p* = 0.666)	
eGFR (mL/min/1.73m^2^)	65.8 ± 34.7	0.012 (−0.001–0.025, *p* = 0.067)	−0.014 (−0.032–0.003, *p* = 0.113)
Urea (mg/dL)	59.5 ± 45.6	−0.003 (−0.013–0.008, *p* = 0.620)	−0.373 (−1.054–0.307, *p* = 0.282)
Potassium (mEq/L)	4.5 ± 0.8	−0.517 (−1.073–0.040, *p* = 0.069)	−0.337 (−0.048–0.004, *p* = 0.093)
Arterial blood glucose (mg/dL)	137 [113; 196]	**−0.006 (−0.012–−0.001, *p* = 0.016)**	**−0.008 (−0.013–− 0.003, *p* = 0.001)**
AST (U/L)	38 [26; 60]	−0.001 (−0.002–0.000, *p* = 0.189)	
ALT (U/L)	28 [17; 49]	−0.002 (−0.006–0.003, *p* = 0.459)	
pH	7.4 ± 0.1	−2.476 (−9.043–4.091, *p* = 0.459)	
pCO_2_ (mmHg)	32.85 ± 8.9	**0.120 (0.064–0.176, *p* < 0.0001)**	0.013 (−0.089–0.115, *p* = 0.797)
pO_2_ (mmHg)	63 [53.4; 74.8]	−0.001 (−0.018–0.015, *p* = 0.879)	
Bicarbonate (mEq/L)	20.98 ± 4.8	**0.207 (0.103–0.310, *p* = 0.0001)**	**0.442 (0.039–0.845, *p* = 0.032)**
Base excess	−2.39 ± 5.4	**0.127 (0.034–0.221, *p* = 0.008)**	−0.241 (−0.521–0.038, *p* = 0.090)

CRP = C-reactive protein; LDH = lactate dehydrogenase; eGFR = estimated glomerular filtration rate; AST = aspartate aminotransferase; ALT = alanine aminotransferase. Hemoglobin, hematocrit, lymphocytes, LDH, eGFR, urea, potassium, venous glucose, pH, pCO_2_, bicarbonate, and base excess are presented as means ± SD. The other variables are presented as medians and IQR. The **bold** values indicate statistical significance.

## Data Availability

The data presented in this study are available upon request from the corresponding authors. The data are not publicly available due to clinical patient information.

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
