# Peer review of "Impact of Hyponatremia on COVID-19-Related Outcomes: A Retrospective Analysis"

_biomedicines, 2024, doi:10.3390/biomedicines12091997_

Round 1
Reviewer 1 Report
Comments and Suggestions for Authors
Good morning to all authors,Analyzing the manuscript (Article) with ID biomedicines-3150828-peer-review-v1, entitled "Impact of hyponatremia on COVID-19-related outcomes: a retrospective analysis" for possible publication in the Journal Biomedicines, Section: Microbiology in Human Health and Disease,
I consider that:
1. The authors proposed a much-discussed topic in today's medical scientific world: The impact of hydroelectrolytic disorders on the evolution of patients with COVID-19.
2. This article follows all the specific instructions of the journal presented in aims and scope, instructions for authors, and other information about the Journal Biomedicines.
3. In Chapter 1 – Introduction: The authors present relevant and important medical information for the chosen topic: The evolution of blood marker levels in COVID-19 patients.
4. In Chapter 2 – Materials and Methods: The authors made a special effort from the beginning of the study and presented very clearly:
Sub-chapter 2.1. – Study population:
- The authors had clear criteria for inclusion and exclusion of patients from this study.
- They had sufficient patients in the study (n=578; evaluated the sodium levels of 558 patients, not available for 20 patients).
- The analyzed patients were admitted to a university hospital (and/or admitted to the ICU).
- The period was chosen to adequately investigate clinical and paraclinical patients with respiratory symptoms (between March 21, 2020 and July 31, 2020).
Sub-chapter 2.2. – Assessment:
- The authors clearly described the medical history (clinical and paraclinical of the patients in the study).
- Tables 1S-4S are suggestive of the study conducted and very clearly presented.
- They had the approval of the Ethics Committee of the Hospital/Institute, thus proving high professional ethics.
Sub-chapter 2.3. – Statistical analyses:
- For the statistical data analysis, the authors used high-performance software: IBM® SPSS (version 18.0).
- The statistical parameters used by the authors are those specific to medical scientific studies: p-value < 0.05; the B value, confidence interval (CI), mean ± SD, and Pearson correlation.
5. In Chapter 3 – Results:
- In the first paragraph you stated: "We evaluated 588 hospitalized patients with COVID-19", in Tables 2 and 3 under Variables n=588!?
- But in Sub-chapter 2.1. – Study population: did you specify "evaluated the sodium levels of 558 patients, not available for 20 patients"!?
- And in Figure 1 and Tables 1 and 4 you specified n=558!?
- In Figure 2 (A-I): How did you make the correlations between the values ​​of serum sodium levels and other biochemical markers? What is the number of patients in the analyzed groups for which you made these correlations?
6. In Chapter 4 - Discussion: According to the bibliography, the authors compared all the results obtained in this study with the results of other studies and the assessments of different authors.
7. All authors have made an equitable contribution to the study.
8. The bibliography chosen by the authors corresponds to the requirements and refers to the subject of the article.
In conclusion:
I ACCEPT with minor revision (clearly specify n=number of investigated patients) for publication in the prestigious Journal Biomedicines (MDPI, ISSN 2227-9059, IF=3,9).

Author Response
We would like to thank the reviewers for their comments, which have certainly contributed to enhancing the manuscript. We have provided a point-by-point response and highlighted the changes in the manuscript in yellow.
Reviewer 1
Good morning to all authors,Analyzing the manuscript (Article) with ID biomedicines-3150828-peer-review-v1, entitled "Impact of hyponatremia on COVID-19-related outcomes: a retrospective analysis" for possible publication in the Journal Biomedicines, Section: Microbiology in Human Health and Disease,
I consider that:
- The authors proposed a much-discussed topic in today's medical scientific world: The impact of hydroelectrolytic disorders on the evolution of patients with COVID-19.
- This article follows all the specific instructions of the journal presented in aims and scope, instructions for authors, and other information about theJournal Biomedicines.
- In Chapter 1 – Introduction: The authors present relevant and important medical information for the chosen topic: The evolution of blood marker levels in COVID-19 patients.
- In Chapter 2 – Materials and Methods: The authors made a special effort from the beginning of the study and presented very clearly:
Sub-chapter 2.1. – Study population:
- The authors had clear criteria for inclusion and exclusion of patients from this study.
- They had sufficient patients in the study (n=578; evaluated the sodium levels of 558 patients, not available for 20 patients).
- The analyzed patients were admitted to a university hospital (and/or admitted to the ICU).
- The period was chosen to adequately investigate clinical and paraclinical patients with respiratory symptoms (between March 21, 2020 and July 31, 2020).
Sub-chapter 2.2. – Assessment:
- The authors clearly described the medical history (clinical and paraclinical of the patients in the study).
- Tables 1S-4S are suggestive of the study conducted and very clearly presented.
- They had the approval of the Ethics Committee of the Hospital/Institute, thus proving high professional ethics.
Sub-chapter 2.3. – Statistical analyses:
- For the statistical data analysis, the authors used high-performance software: IBM® SPSS (version 18.0).
- The statistical parameters used by the authors are those specific to medical scientific studies: p-value < 0.05; the B value, confidence interval (CI), mean ± SD, and Pearson correlation.
- In Chapter 3 – Results:
- In the first paragraph you stated: "We evaluated 588 hospitalized patients with COVID-19", in Tables 2 and 3 under Variables n=588!?
- But in Sub-chapter 2.1. – Study population: did you specify "evaluated the sodium levels of 558 patients, not available for 20 patients"!?
- And in Figure 1 and Tables 1 and 4 you specified n=558!?
Response: Thank you for bringing this information to our attention. As we stated on Page 2, lines 47-49, and page 3, lines 2-4: “A total of 578 patients were hospitalized. Exclusion criteria were patients under 18 years old and the absence of sodium measurement within the first 24 hours of hospital admission and/or ICU admission. As a result, sodium measurements were not available for 20 patients at the time of hospital/ICU admission, so they were excluded from the analysis. Consequently, 558 patients were included in the study.” We have therefore corrected the numbers in Tables 2 and 3, changing 588 to 558.
- In Figure 2 (A-I): How did you make the correlations between the values ​​of serum sodium levels and other biochemical markers? What is the number of patients in the analyzed groups for which you made these correlations?
Response: We added the notation “N=558” at the end of the legend.
- In Chapter 4 - Discussion: According to the bibliography, the authors compared all the results obtained in this study with the results of other studies and the assessments of different authors.
- All authors have made an equitable contribution to the study.
- The bibliography chosen by the authors corresponds to the requirements and refers to the subject of the article.
In conclusion:
I ACCEPT with minor revision (clearly specify n=number of investigated patients) for publication in the prestigious Journal Biomedicines (MDPI, ISSN 2227-9059, IF=3,9).
Response: Thank you for your comments and for supporting our study.
Reviewer 2 Report
Comments and Suggestions for Authors
This retrospective study explored the relationship between demographic data, clinical symptoms, laboratory tests and hyponatremia during COVID-19, as many studies have shown that hyponatremia is associated with poor prognosis of COVID-19 patients. The study design was rigorous, the analysis method was selected properly, and the correlation between related indicators and hyponatremia was relatively fully analyzed in the discussion section, which I believe is of great significance for the prediction of hyponatremia and the stratification of COVID-19 patients. However, we would also like to see this study's reference and guiding significance for the future and hope to add this part moderately in the discussion section. Second, it is desirable to refine the inclusion and exclusion criteria of the study and supplement the use of medications (e.g., diuretics and corticosteroids, which can affect patients' electrolytes).
Comments on the Quality of English LanguageSome of the language needs to be modified to suit the professional audience.
Author Response
We would like to thank the reviewers for their comments, which have certainly contributed to enhancing the manuscript. We have provided a point-by-point response and highlighted the changes in the manuscript in yellow.
Reviewer 2
This retrospective study explored the relationship between demographic data, clinical symptoms, laboratory tests and hyponatremia during COVID-19, as many studies have shown that hyponatremia is associated with poor prognosis of COVID-19 patients. The study design was rigorous, the analysis method was selected properly, and the correlation between related indicators and hyponatremia was relatively fully analyzed in the discussion section, which I believe is of great significance for the prediction of hyponatremia and the stratification of COVID-19 patients.
1) However, we would also like to see this study's reference and guiding significance for the future and hope to add this part moderately in the discussion section.
Response: Thank you for your suggestion. We expanded the last paragraph of the manuscript, as follow: “In conclusion, clinical and demographic parameters can predict sodium values at admission. Early identification of factors that predict COVID-19 progression aids in therapeutic decision-making and patient flow management. Identifying predictors of poor outcomes in COVID-19 is crucial for effective risk stratification. Additionally, our study provides further biological insights into disease progression. Thus, our findings contribute to resource planning and allocation and offer valuable data for developing machine learning models in future pandemic waves of COVID-19 and other respiratory viruses in general.”
2) Second, it is desirable to refine the inclusion and exclusion criteria of the study and supplement the use of medications (e.g., diuretics and corticosteroids, which can affect patients' electrolytes).
Response: Thank you for your comments. We added the following sentences to the “Limitations’ section: “The use of other medications that could potentially interfere with electrolyte balance, such as diuretics and corticosteroids, as well as the assessment of adrenal and thyroid function, were not available.”
Sincerely,
Érika B Rangel, MD, PhD
Reviewer 3 Report
Comments and Suggestions for Authors
In this manuscript, Pedro Maciel de Toledo Piza and colleagues explore the potential association between hyponatremia and a variety of demographic, clinical and laboratory parameters, by evaluating the sodium levels of a cohort of 558 inpatients recruited at one center during the very first wave of COVID-19 pandemic. The methods used throughout the paper are appropriate, and the data and results obtained are very clear. Moreover, what stands out most is the authors’ comprehensive analysis of each parameter and their discussion of its potential mechanism and outcomes in COVID-19 patient. However, there are several technical parts that need to be revised before publication.
1. Please verify if it is 588 or 558 in the penultimate line of page 3.
2. Authors should clearly point out whether the lower sodium levels are negatively or positively associated with COVID-19 outcomes/demographic data with necessary values. Meanwhile, the significance of these associations should be discussed here with corresponding p values.
3. Author should also analyze the implications when the p values for ICU and IMV in multivariate analysis are greater than 0.05, while their corresponding values in Univariate analysis are smaller than 0.05.
4. Authors should indicate beneath the tables why some numbers are highlighted in bold in table 1-4.
5. ‘Univariate analysis’ should be indicated in the first half of paragraph that describes Table 4 and authors are supposed to explain why Neutrophil-to-lymphocyte is not mentioned here under the univariate analysis
6. r values should be discussed when describing figure 2, given all of them are extremely small.
Author Response
We would like to thank the reviewers for their comments, which have certainly contributed to enhancing the manuscript. We have provided a point-by-point response and highlighted the changes in the manuscript in yellow.
In this manuscript, Pedro Maciel de Toledo Piza and colleagues explore the potential association between hyponatremia and a variety of demographic, clinical and laboratory parameters, by evaluating the sodium levels of a cohort of 558 inpatients recruited at one center during the very first wave of COVID-19 pandemic. The methods used throughout the paper are appropriate, and the data and results obtained are very clear. Moreover, what stands out most is the authors’ comprehensive analysis of each parameter and their discussion of its potential mechanism and outcomes in COVID-19 patient. However, there are several technical parts that need to be revised before publication.
- Please verify if it is 588 or 558 in the penultimate line of page 3.
Response: Thank you for bringing this information to our attention. As we stated on Page 2, lines 47-49, and page 3, lines 2-4: “A total of 578 patients were hospitalized. Exclusion criteria were patients under 18 years old and the absence of sodium measurement within the first 24 hours of hospital admission and/or ICU admission. As a result, sodium measurements were not available for 20 patients at the time of hospital/ICU admission, so they were excluded from the analysis. Consequently, 558 patients were included in the study.” We have therefore corrected the numbers in Tables 2 and 3, changing 588 to 558.
- Authors should clearly point out whether the lower sodium levels are negatively or positively associated with COVID-19 outcomes/demographic data with necessary values. Meanwhile, the significance of these associations should be discussed here with corresponding p values.
Response: Thank you for your suggestion. We have performed amendments to the manuscript accordingly.
- Author should also analyze the implications when the p values for ICU and IMV in multivariate analysis are greater than 0.05, while their corresponding values in Univariate analysis are smaller than 0.05.
Response: Thank you for your comments. In the univariate analysis, we found a p-value less than 0.05 for both outcomes. Since almost two-thirds of the patients who required ICU transfer also required IMV, there was likely an interaction between the two variables.
- Authors should indicate beneath the tables why some numbers are highlighted in bold in table 1-4.
Response: As suggested, we have added the sentence “The bold values indicate statistical significance” beneath the tables.
- ‘Univariate analysis’ should be indicated in the first half of paragraph that describes Table 4 and authors are supposed to explain why Neutrophil-to-lymphocyte is not mentioned here under the univariate analysis.
Response: Thank you for your suggestion. We have performed amendments to the manuscript accordingly.
- r values should be discussed when describing figure 2, given all of them are extremely small.
Response: We agree with the reviewer that low Pearson ? values were obtained, indicating a weak correlation. A comprehensive overview of Pearson's correlation coefficient shows that a strong correlation is indicated by values between 0.5 and 1, a moderate correlation by values between 0.30 and 0.49, and a weak correlation by values below 0.29. Possible Reasons for a Low r Value: a) Non-Linear Relationships: The relationship between the variables might be non-linear. Pearson’s r only captures linear relationships, so a low r value does not mean there is no relationship at all; it could be non-linear. b) Presence of Outliers: Outliers can influence the correlation coefficient, potentially leading to a low r value. c) Range of Data: If the data range is narrow or limited, it might result in a low r value. d) Sample Size: A small sample size might not provide a reliable estimate of the correlation. We have added this information to the 'Limitations' section."
Sincerely,
Érika B Rangel, MD, PhD